# A non-canonical role for the autophagy machinery in anti-retroviral signaling mediated by TRIM5α

**Bhaskar Saha**[1], **Devon Chisholm**[1], **Alison M. Kell**[1], **Michael A. Mandell**[1,2] *

**1** Department of Molecular Genetics and Microbiology, University of New Mexico Health Sciences Center, Albuquerque, New Mexico, United States of America, **2** Autophagy, Inflammation and Metabolism Center of Biomedical Research Excellence, University of New Mexico Health Sciences Center, Albuquerque, New Mexico, United States of America

* mmandell@salud.unm.edu

**Data Availability Statement:** All relevant data are within the manuscript and its Supporting Information files.

**Funding:** M.A.M. was supported by NIH grant 1P20GM121176 (NIGMS) and 1R21AI131964

## Abstract

TRIM5α is a key cross-species barrier to retroviral infection, with certain TRIM5 alleles conferring increased risk of HIV-1 infection in humans. TRIM5α is best known as a species-specific restriction factor that directly inhibits the viral life cycle. Additionally, it is also a pattern-recognition receptor (PRR) that activates inflammatory signaling. How TRIM5α carries out its multi-faceted actions in antiviral defense remains incompletely understood. Here, we show that proteins required for autophagy, a cellular self-digestion pathway, play an important role in TRIM5α's function as a PRR. Genetic depletion of proteins involved in all stages of the autophagy pathway prevented TRIM5α-driven expression of NF-κB and AP1 responsive genes. One of these genes is the preeminent antiviral cytokine interferon β (IFN-β), whose TRIM5-dependent expression was lost in cells lacking the autophagy proteins ATG7, BECN1, and ULK1. Moreover, we found that the ability of TRIM5α to stimulate IFN-β expression in response to recognition of a TRIM5α-restricted HIV-1 capsid mutant (P90A) was abrogated in cells lacking autophagy factors. Stimulation of human macrophage-like cells with the P90A virus protected them against subsequent infection with an otherwise resistant wild type HIV-1 in a manner requiring TRIM5α, BECN1, and ULK1. Mechanistically, TRIM5α was attenuated in its ability to activate the kinase TAK1 in autophagy deficient cells, and both BECN1 and ATG7 contributed to the assembly of TRIM5α-TAK1 complexes. These data demonstrate a non-canonical role for the autophagy machinery in assembling antiviral signaling complexes and in establishing a TRIM5α-dependent antiviral state.

## Author summary

TRIM5α is an antiretroviral protein that employs multiple mechanisms to protect cells against infection. Previous studies have linked TRIM5α to autophagy, a cytoplasmic quality control pathway with numerous roles in immunity, raising the possibility that TRIM5α engages autophagy in antiviral defense. This concept has been controversial, since TRIM5α's best-known role as a directly acting antiretroviral effector is autophagy

(NIAID). A.M.K. was supported by NIH grant 1K22AI141680-01A1 (NIAID). The funders had no role in study design, data collection and analysis, decision to publish, or preparation of the manuscript.

**Competing interests:** The authors have declared that no competing interests exist.

independent. However, retroviral restriction is only one aspect of TRIM5α function. We demonstrate that autophagy is crucial to another TRIM5α action: its role as a pattern-recognition receptor. We show that autophagy machinery is required for TRIM5α to transduce antiviral signaling and to establish an antiviral state. Our data indicate that autophagy provides TRIM5α with a platform upon which to activate antiviral responses.

## Introduction

The selective pressure imposed by retroviral infection has shaped the human genome and has driven the evolution of proteins that function to protect host cells against retroviral infection [1]. These proteins are referred to as 'restriction factors' [2]. One of the first restriction factors to be identified was tripartite motif-containing protein 5, isoform α (TRIM5 hereafter), which was shown to potently block the ability of HIV-1 to infect cells from certain species of old-world monkeys (e.g. Rhesus macaques) [3,4]. The ability of TRIM5 to restrict incoming retroviruses depends on whether or not the C-terminal PRY/SPRY domain of TRIM5 can recognize and bind to the retroviral capsid proteins in the context of an intact core structure. These TRIM5-capsid interactions show a high degree of both host and viral species specificity, with human TRIM5 historically being considered unable to bind and restrict HIV-1 despite its ability to efficiently block infection by other retroviruses [3–8]. Interestingly, despite TRIM5's relative inability to protect human cells from HIV-1 infection *in vitro*, human genetic studies have indicated that certain TRIM5 alleles confer increased risk of HIV-1 infection in people [9–11]. This implies that TRIM5 likely plays additional roles in antiretroviral defense beyond its actions in capsid-specific restriction (CSR).

These expanded functions of TRIM5 have started to emerge. For instance, TRIM5 can act as a pattern-recognition receptor for retroviral capsid, triggering the expression of NF-κB- and AP1-regulated immune genes (e.g. interferon α/β, IL-6) [12–15]. At its N terminus, TRIM5 encodes a RING E3 ubiquitin ligase domain. TRIM5 assembly on retroviral core structures potentiates the enzymatic activity of the RING domain [12,13]. This in turn enables the ubiquitin-sensing kinase and TRIM5-interacting protein TAK1 (TGFβ-activated kinase 1; MAP3K7) [12], a key component in multiple pathogen-sensing and cytokine-signaling pathways. TRIM5's ability to stimulate immune signaling through TAK1 is genetically separable from its actions in CSR as only the former process requires TRIM5 enzymatic activity [13].

Additionally, we and others have reported multiple linkages between TRIM5 and the macroautophagy pathway [16–18]. Macroautophagy (autophagy hereafter) is a cellular mechanism of self-digestion in which cytoplasmic contents are sequestered within a vesicle (the autophagosome) which subsequently fuses with lysosomes [19]. Under conditions of nutrient limitation, autophagy can promote the bulk degradation of cytoplasm to provide the cell with molecules needed for biosynthesis. Alternatively, the autophagy machinery can preferentially target specific substrates for degradation in a process referred to as 'selective autophagy'. Many studies have demonstrated that either the autophagy pathway as a whole or individual components of the autophagy machinery have a broad array of physiological roles in humans [20]; with controlling innate immune responses and antiviral defense being prominent examples [21]. On a molecular level, the autophagy pathway can be conceptually sub-divided into three coordinated steps generally classified as autophagy initiation, autophagosome formation, and autophagosome-lysosome fusion [19,20]. These steps are carried out by ~20 evolutionarily conserved 'core' autophagy proteins, with extensive crosstalk and coordination between the actions of proteins functioning in multiple steps. In the initiation stage, autophagy-inducing

signals converge on the most upstream autophagy regulator ULK1. Once activated, ULK1 phosphorylates and activates a number of downstream targets including the BECN1/hVPS34 complex 1 which includes the proteins BECN1, AMBRA1, and ATG14. This triggers the generation of phosphatidylinositol-3-phosphate (PI3P) on membranes at the autophagy initiation site. PI3P subsequently recruits proteins involved in forming the autophagosome membrane. Among these are ATG7 and ATG16L, which help catalyze the lipidation of the mammalian paralogues of yeast Atg8 (mAtg8s; LC3A-C and the three GABARAPs). Once lipidated, mAtg8 proteins associate with the autophagosome membrane and are thought to play roles in providing its shape and in recruiting proteins involved in autophagosome-lysosome fusion. Additionally, mAtg8s interact with autophagic cargo receptors (e.g. p62/Sequestosome 1), which act as bridging factors between selective autophagy substrates and the autophagosomal membrane. The maturation of the autophagosome, culminating in fusion with lysosomes, involves the actions of the BECN1/hVPS34 complex 2 (BECN1 and UVRAG), Rab7, and several membrane fusion proteins (e.g. VAMP8 and SNAP29). TRIM family proteins, including TRIM5, are increasingly recognized for modulating the actions and activities of these core autophagy factors [22].

TRIM5 has been shown to interact with proteins required for the upstream initiation of autophagy (ULK1, BECN1, ATG14, and AMBRA1), with mAtg8 conversion machinery (ATG5 and ATG16L), with all members of the mAtg8 protein family, with the autophagosome maturation factor UVRAG, and with the selective autophagy receptor p62 [16,18,23,24]. Depletion of TRIM5 attenuates autophagic responses to mTOR inhibition [16] and to viral infection [25]. These multiple connections between TRIM5 and autophagy raise the question of whether TRIM5 may involve the autophagy machinery in its multifaceted defense of cells from retroviral infection. While a role for autophagy in TRIM5-dependent CSR has been excluded [26], the possibility that autophagy underlies some of TRIM5's other actions remains intriguing.

Here we show that the autophagy machinery strongly contributes to the ability of TRIM5 to act as a pro-inflammatory signaling molecule and for it to establish a cellular antiviral state in response to retroviral capsid detection. Our studies suggest that autophagy proteins are required to provide a scaffold for the interaction between TRIM5 and the kinase TAK1. Autophagy factor depletion attenuates the ability of TRIM5 to drive TAK1 activation and downstream signaling. Collectively, these studies suggest a novel, non-degradative role of the autophagy machinery in assembling active signaling complexes and uncover a role for autophagy in TRIM5-mediated antiviral responses.

## Results

### Autophagy machinery is required for TRIM5-driven NF-κB and AP1 trans-activation

TRIM5 over-expression in HEK293T cells drives activation of NF-κB and AP1 reporters in a manner requiring the kinase TAK1 and the E2 ubiquitin conjugating enzyme UBC13/UBE2N (Fig 1A) [12,13]. We used a previously published dual-luciferase reporter assay in HEK293T cells to determine whether autophagy factors contributed to this process. In control experiments, we found that transient expression of both N-terminally and C-terminally tagged TRIM5 enhanced the relative expression of AP1-driven luciferase relative to cells expressing the tags alone (S1A Fig). This assay was set up so that additional TRIM5 expression would increase transcription factor activation [13]. The effect of TRIM5 over-expression on AP1 was comparable to that of expressing GFP-tagged TAK1. Expression of a dominant-negative kinase-dead mutant of TAK1 (K63W) did not activate, and in fact reduced, AP1 signaling. As

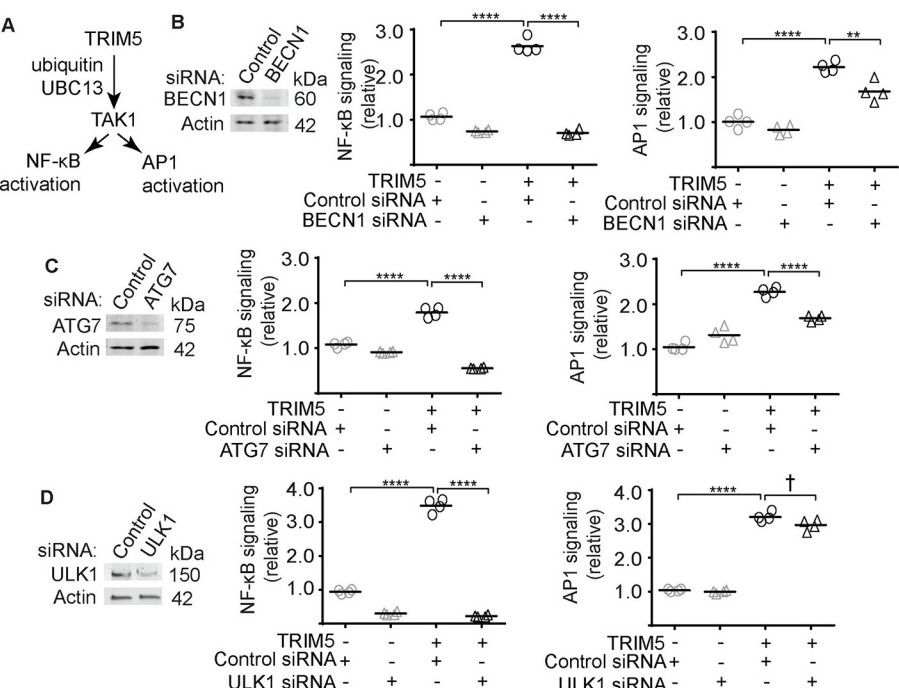

**Fig 1. Requirement for core autophagy machinery in TRIM5-based activation of inflammatory transcription factors NF-κB and AP1.** (**A**) Schematic of TRIM5-TAK1 signaling axis. (**B-D**) HEK293T cells were transfected with non-targeting or the indicated siRNA one day prior to transfection with plasmids encoding firefly luciferase under the control of NF-κB- or AP1-responsive promoters, constitutively active *Renilla* luciferase, and either TRIM5-APEX2 or APEX2 alone. 40–48h after transfection, samples were harvested, and luciferase values determined. Immunoblots to the left indicate knockdown efficiency. Plots show the relative signaling, which is determined by dividing a sample's firefly luciferase signal by its *Renilla* luciferase signal. All values are then normalized to the mean of the TRIM5-negative control siRNA-treated sample. Data, each data point represents an independent biological replicate. **, P < 0.01; ****, P < 0.0001; †, not significant by ANOVA.

expected, the ability of TRIM5 to activate signaling through both NF-κB and AP1 was abrogated following siRNA-mediated knockdown of UBC13 (S1B Fig). Surprisingly, knockdown of the autophagy factors BECN1 and ATG7 eliminated the ability of TRIM5 to activate NF-κB reporter expression and substantially reduced its ability to activate AP1 (Fig 1B and 1C). Knockdown of ULK1 also abrogated TRIM5-driven NF-κB activation but did not significantly impact AP1 activation (Fig 1D). To extend these studies, we generated HEK293T cell lines in which we had knocked out ULK1, BECN1, and ATG7 by CRISPR/Cas9 (Fig 2A). We also generated an accompanying cell line expressing Cas9 and a non-targeting gRNA for use as a negative control. TRIM5 expression in the control cells induced NF-κB and AP1 signaling by 2–4 fold. The effect of TRIM5 expression on NF-κB reporter activation was lost in the autophagy factor knockout cells (Fig 2B). As was the case with the siRNA experiments, AP1 reporter expression was still induced by TRIM5 in the BECN1 and ATG7 knockout cells, albeit substantially less that what was seen in the control cells. Notably, ULK1 knockout cells were also reduced in their ability to activate AP1-dependent gene expression in response to TRIM5 expression (Fig 2C). In contrast to what is seen with TRIM5, autophagy factor knockout did not prevent NF-κB reporter activation in cells over-expressing TAK1 (Fig 2D), indicating that autophagy factors are specifically required for TRIM5 signaling but not at points further downstream. These data demonstrate that early-acting autophagy factors (ULK1 and BECN1) and the mAtg8 conjugation machinery (ATG7) are required for TRIM5-driven signaling.

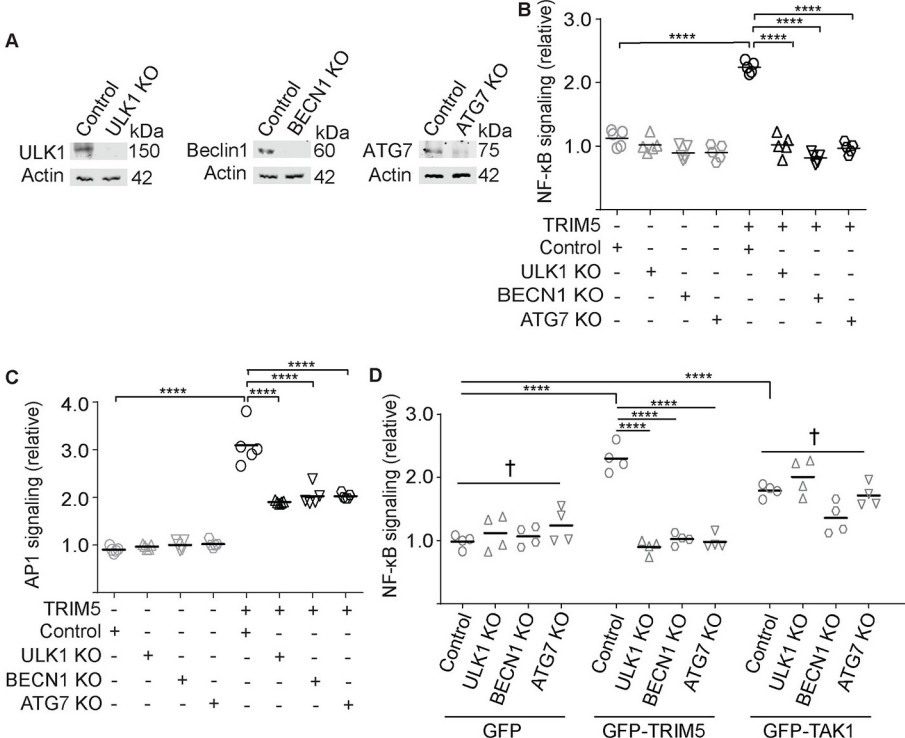

**Fig 2. Knockout of core autophagy machinery attenuates TRIM5 signaling at the level of TAK1.** (**A**) Immunoblots showing knockouts of ULK1, BECN1, and ATG7 from CRISPR/Cas9-generated HEK293T cells. Control lines were generated expressing Cas9 and non-targeting guide RNAs. (**B,C**) Control or autophagy factor knockout HEK293T cells were transfected with plasmids encoding firefly luciferase under the control of NF-κB- or AP1-responsive promoters, constitutively active *Renilla* luciferase, and either TRIM5-APEX2 or APEX2 alone and luciferase activity determined as mentioned previously. (**D**) Control and knockout HEK293T cells were transfected with NF-κB reporter plasmid and GFP-TRIM5, GFP-TAK1, or GFP alone, with NF-κB signaling as determined above. Data, each data point represents an independent biological replicate. ****, P < 0.0001; †, not significant by ANOVA.

We next asked what other modules of the autophagy machinery are required for TRIM5-based NF-κB and AP1 transcription factor activation. The results of the ATG7 experiments above indicated a role for mAtg8 proteins, which we confirmed following GABARAP knockdown (Figs 3A and S1C). The same effect was seen for the autophagy receptor p62 (Figs 3B and S1C), but this could potentially be related to the previously described effects of p62 knockdown on TRIM5 expression or stability [24]. The expression of many autophagy factors are under the control of the transcription factor TFEB, which is a master regulator of lysosome biogenesis [27]. In accordance with the datasets above, TFEB knockdown also strongly impacted the ability of TRIM5 expression to drive NF-κB and AP1 activation in this assay (Figs 3C and S1C). Finally, we tested whether autophagosome closure or autophagosome-lysosome fusion were required for TRIM5-signaling by knocking down the ESCRT component CMHP2A [28], the autophagosome-associated (t)-SNARE SNAP29, and the lysosomal (v)-SNARE VAMP8, and found that these proteins were also essential for TRIM5-based signaling (Figs 3D, 3E and S1C). Taken together, these data demonstrate that proteins required for autophagy initiation, autophagosome membrane formation and closure, substrate selectivity, and lysosomal fusion all play crucial roles in TRIM5-dependent signaling in this reporter cell assay. Experiments using the lysosomal protease inhibitors e64D and pepstatin A indicated that the autophagy machinery may be playing a non-degradative role in enabling TRIM5

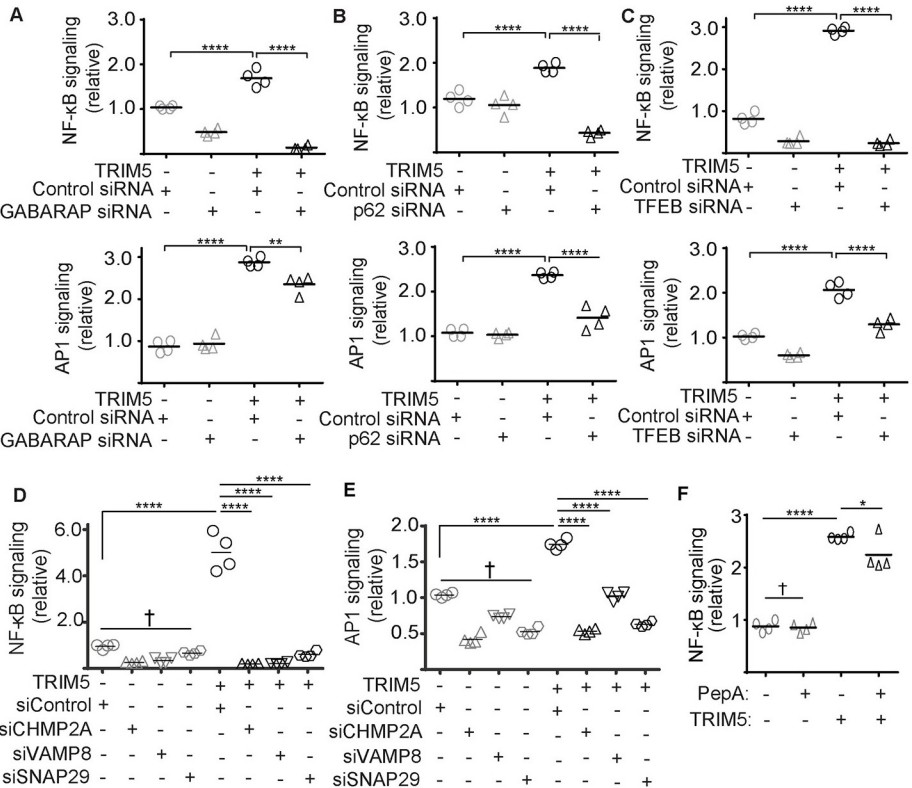

**Fig 3. Multiple components of autophagy/lysosome system are involved in TRIM5 signaling.** (A-E) The effect of the indicated knockdowns on TRIM5-mediated NF-κB and AP1 activation in a dual luciferase reporter assay system. Immunoblots demonstrating knockdown efficiency are shown in S1C Fig. (F) HEK293T cells were transfected as above and treated or not with the protease inhibitors e64d and pepstatin A (PepA) for 48 h prior to lysis. Both inhibitors were use at a concentration of 10 μM. Luciferase activity determined as above. Each data point represents an independent biological replicate. *, P < 0.05; **, P < 0.01; ****, P < 0.0001; †, not significant by ANOVA.

signaling, since e64d and pepstatin A only modestly diminished the impact of TRIM5 expression on NF-κB activity (Fig 3F).

## Autophagy factors are required for TRIM5-driven immune gene expression in macrophages

We next tested whether autophagy factor knockdown could impact TRIM5-driven gene expression in PMA-differentiated THP1 macrophages, as macrophages are one of the first cell types that HIV-1 is likely to encounter following transmission. To this end, we generated polyclonal cell lines stably transduced with C-terminally tagged rhesus or human TRIM5 or with the protein tag alone (S2A Fig). These cells showed the expected differential sensitivity to transduction with GFP-encoding HIV-1 pseudovirions, with the rhesus TRIM5-expressing cell line being strongly protected relative to the other two cell lines (Fig 4A). In qPCR experiments, we found that both the human TRIM5 and rhesus TRIM5 cell lines showed elevated expression of the preeminent antiviral cytokine interferon β relative to control THP1 cells under homeostatic conditions. However, siRNA-mediated knockdown of the autophagy factors BECN1 or ATG7 in the TRIM5-overexpressing cell lines reduced the expression of *IFNB1* mRNA to levels comparable to that seen in the control cells (Figs 4B, S2C and S2D). These experiments also demonstrated that the expression of the inflammasome component *NLRP1*

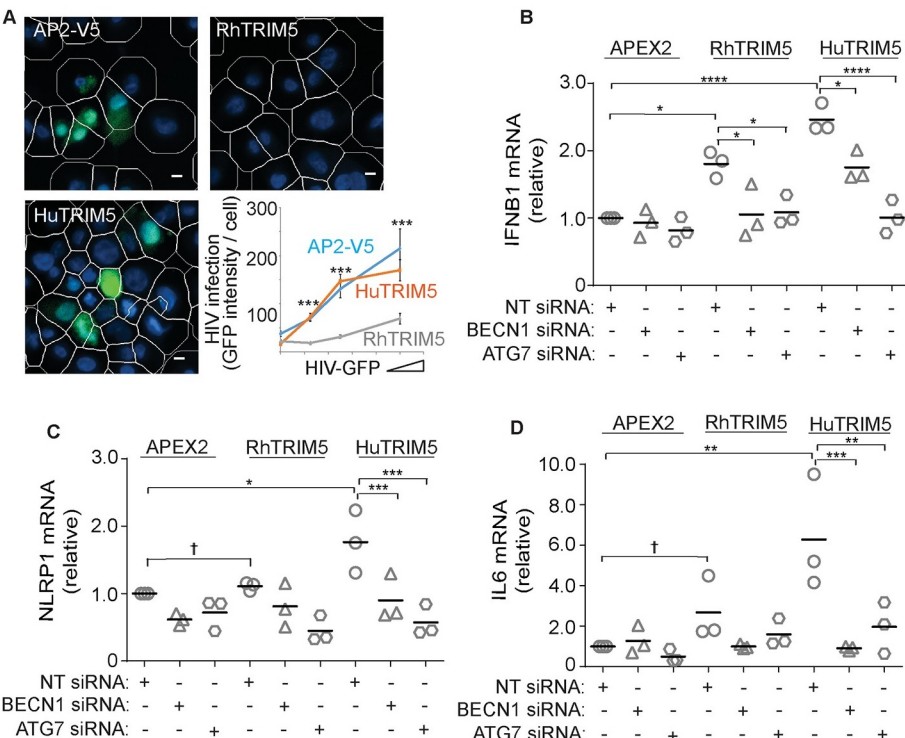

**Fig 4. Autophagy factors are required for TRIM5-driven inflammatory gene expression in macrophages.** (**A**) Differentiated THP-1 cell lines stably expressing APEX2-V5 alone or APEX2-V5 tagged TRIM5 (human or rhesus) were infected with HIV-GFP and the percentage of infected (GFP-positive) cells was determined by high content imaging. White mask, automatically identified cell boundaries. Scale bar, 50 μm. Plot, the mean above-background GFP intensity per cell (a measure of transduction efficiency) following infection of increasing amounts of HIV-GFP. N = 5 independent biological replicates, >500 cells analyzed per replicate. (**B-D**) Quantitative PCR analysis of THP-1 macrophage cells stably transduced with lentiviruses expressing rhesus or human TRIM5 fused to APEX2 or APEX2 alone and subjected to the indicated siRNA-mediated knockdowns. The abundance of *IFNB1* (B), *NLRP1* (C), and *IL6* (D) were determined relative to the abundance of 18S ribosomal RNA. Quantitative PCR analysis of knockdown efficiency is shown in S2C and S2D Fig. Data, each data point represents an independent biological replicate; *, P < 0.05; **, P < 0.01; ***, P < 0.001, ****, P < 0.0001; †, not significant by ANOVA.

and the cytokines *IL6* and *IL10* were elevated at the mRNA level in the THP1 cells over-expressing human TRIM5 following transfection with non-targeting siRNA but not following transfection with BECN1 or ATG7 siRNA (Figs 4C, 4D and S2B). These data recapitulate our findings from the reporter cell assays and demonstrate a requirement for autophagy factors in inflammatory signaling driven by TRIM5 expression in macrophages.

## TRIM5 actions as a pattern-recognition receptor require autophagy factors

TRIM5 can stimulate innate immune signaling in response to its recognition of restriction-sensitive retroviral cores [12–14]. Our results showing a requirement for autophagy factors in activation of these same immune pathways in response to TRIM5 over-expression suggested that autophagy factors may also contribute to TRIM5-dependent signaling in response to retroviral capsid. For these studies, we used an HIV-1 mutant with a single amino acid substitution in the capsid protein (P90A) that was recently shown to be strongly restricted by human TRIM5 in primary immune cells [29,30]. Additionally, P90A was previously shown to activate innate immune signaling and interferon expression [31], but this activity was not connected to TRIM5. We confirmed that the P90A virus is restricted by TRIM5 in THP-1 cells, as TRIM5

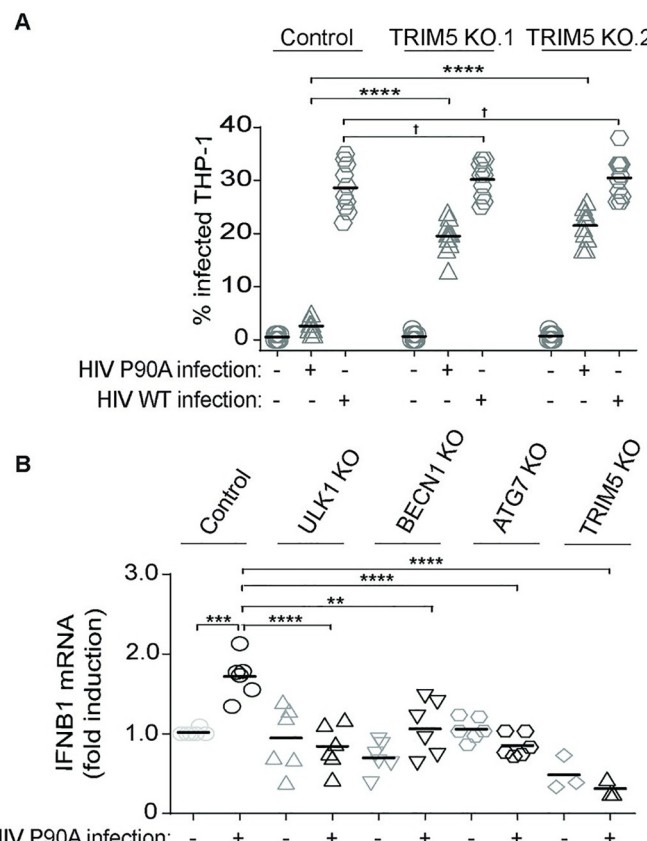

**Fig 5. TRIM5 actions as a pattern-recognition receptor require autophagy factors.** (**A**) Effect of TRIM5 knockout on the ability of HIV-1 CA P90A and WT HIV-1 to infect THP-1 macrophages. Control or TRIM5 KO cells (lines 1 and 2) were PMA-differentiated and then infected with single-cycle fluorescent protein-expressing pseudovirions for two days (CrFK MOI = 1). The percentage of cells showing red fluorescence (P90A) or green fluorescence (HIV WT) was determined by high content imaging 2 days post infection. Each data point represents the results from an individual well of a 96-well plate (>2000 cells analyzed per well). TRIM5 KO line 2 was used for subsequent experiments. (**B**) The effect of HIV-1 P90A infection on the expression of inflammatory gene *IFNB1* in THP-1 macrophages expressing or not autophagy factors and TRIM5. THP-1 cell lines knocked out for the autophagy factors ULK1, BECN1, ATG7 or TRIM5 (or control) were PMA-differentiated and then exposed to HIV-1 CA P90A for two hours (CrFK MOI = 3) prior to RNA isolation and qPCR analysis. 18S ribosomal RNA was used as a normalization control. Data, each data point represents an independent biological replicate; **, P < 0.01; ***, P < 0.001, ****, P < 0.0001; †, not significant by ANOVA.

knockout increase P90A transduction efficiency by ~10 fold (Figs 5A and S3A). Transduction efficiency of WT HIV was not impacted by TRIM5 knockout, and the WT virus was still slightly more infectious than P90A in the TRIM5 KO cells. For these experiments, MOI was determined in permissive TRIM5-null CrFK cells, and an equal CrFK MOI of WT and P90A virus was used. As expected, TRIM5 knockout also increased the ability of N-tropic murine leukemia virus (N-MLV) to infect THP-1 macrophages (S3B Fig). When THP-1 macrophages are exposed to equal MOI of WT or P90A HIV, only P90A induced the expression of *IFNB1* mRNA (S3C Fig). Together, these results are consistent with HIV-1 CA P90A infection stimulating TRIM5-dependent inflammatory gene expression. We next generated ULK1, BECN1, and ATG7 THP-1 cell lines (S3D Fig) to test whether HIV-1 CA P90A infection could similarly induce the expression of pro-inflammatory genes in cells lacking autophagy. Whereas infection of control THP-1 macrophages with HIV-1 CA P90A increases the expression of *IFNB1*, this effect is not seen in ULK1, BECN1, or ATG7 knockout THP-1 cells (Fig 5B). In

these experiments, TRIM5 knockout THP-1 cells expressed reduced *IFNB1* under basal conditions and *IFNB1* expression was not increased in response to HIV-1 CA P90A infection. These data suggest that these autophagy-related proteins are critical to TRIM5-dependent cellular responses to retroviral capsid recognition.

## TRIM5 establishes a cellular antiviral state in a manner requiring autophagy factors

In the experiments described above, we saw that HIV-1 CA P90A induced the expression of antiviral interferon β in an autophagy factor-dependent manner. We next asked whether this effect was sufficient to affect the outcome of infection with TRIM5-resistant viruses. To do this, we devised a scheme of sequential infections in which THP-1 macrophages are "primed" with human TRIM5-restricted single-cycle viruses prior to being "challenged" by GFP-expressing HIV-1 pseudovirions with wild type (TRIM5-resistant) capsid. The number of HIV-1 infected (GFP-positive) cells is then determined by high content imaging (Fig 6A). We found that priming of THP-1 cells with either HIV-1 CA P90A (Figs 6B, 6C and S4A) or with N-MLV (Figs 6D and S4B) reduced the ability of TRIM5-resistant WT HIV-1 to infect THP-1 macrophages by 3–4 fold relative to what was seen in unprimed THP-1. The effect of priming correlated with the species-specific ability of TRIM5 to restrict the priming virus, as priming with an equal CrFK MOI of a WT capsid-bearing virus that was otherwise identical to HIV-1 CA P90A was significantly less efficient in restricting WT HIV-1 infection of THP1 cells (Figs 6E and S4C). Furthermore, the HIV-restrictive effect of priming was substantially reduced (albeit not eliminated) in TRIM5 knockout THP-1 cells (Fig 6F) and was restored following TRIM5 add-back (Figs 6G and S4D), thus establishing that the effect is at least partially TRIM5-dependent. These results mirror those reported by Merindol et al [14], who found that prior exposure of macrophages to TRIM5-restricted HIV-1 isolates from an elite controller population conferred protection against infection by wild type HIV-1 in a manner requiring TRIM5, UBC13, and TAK1. Since the ability of human TRIM5 to restrict HIV-1 in THP-1 cells is only effective secondarily to priming with a restricted virus, we refer to this effect as secondary TRIM5 restriction (STR). TRIM5-dependent STR is not specific to cell type, as P90A priming of the hepatoma cell line Huh7 protected those cells from transduction with WT HIV pseudovirus (S4E Fig). In addition to restricting HIV, STR can also reduce Sendai virus infection of primed THP-1 macrophages comparably to pretreatment of the cells with the TLR-3 ligand poly(I:C) in a TRIM5-dependent manner (Fig 6H). As expected, TRIM5 knockout did not impact the ability of poly(I:C) treatment to restrict Sendai virus. Together, these data show that TRIM5 can establish an antiviral state in response to retroviral capsid recognition.

We next tested whether priming with HIV-1 CA P90A could attenuate WT HIV-1 infection in ULK1 and BECN1 knockout THP-1 cells. We did not pursue these experiments with the ATG7 knockout THP-1 cells since this cell line showed enhanced resistance to HIV-1 transduction even in the absence of priming (S4F Fig). The results from these experiments were consistent with what we have reported above regarding the role of autophagy factors in TRIM5-dependent signaling: the effect of HIV-1 CA P90A priming was largely abrogated in autophagy factor knockout cells. The magnitude of this effect was comparable to that of TRIM5 knockout (Fig 7A–7C). Furthermore, chemical inhibition of the NF-κB and AP1 pathways, or of autophagy, with the compounds BAY11-7085, SP600125, and 3-methyladenine respectively attenuated the anti-HIV effect of P90A priming (Figs 7D and S4G). Proteasome inhibition with MG132 did not impact STR (S4H Fig). Taken together, these data demonstrate that activation of TRIM5 by a restricted retrovirus can establish an antiviral state that can protect against otherwise TRIM5-insensitive viruses. They furthermore demonstrate that at least

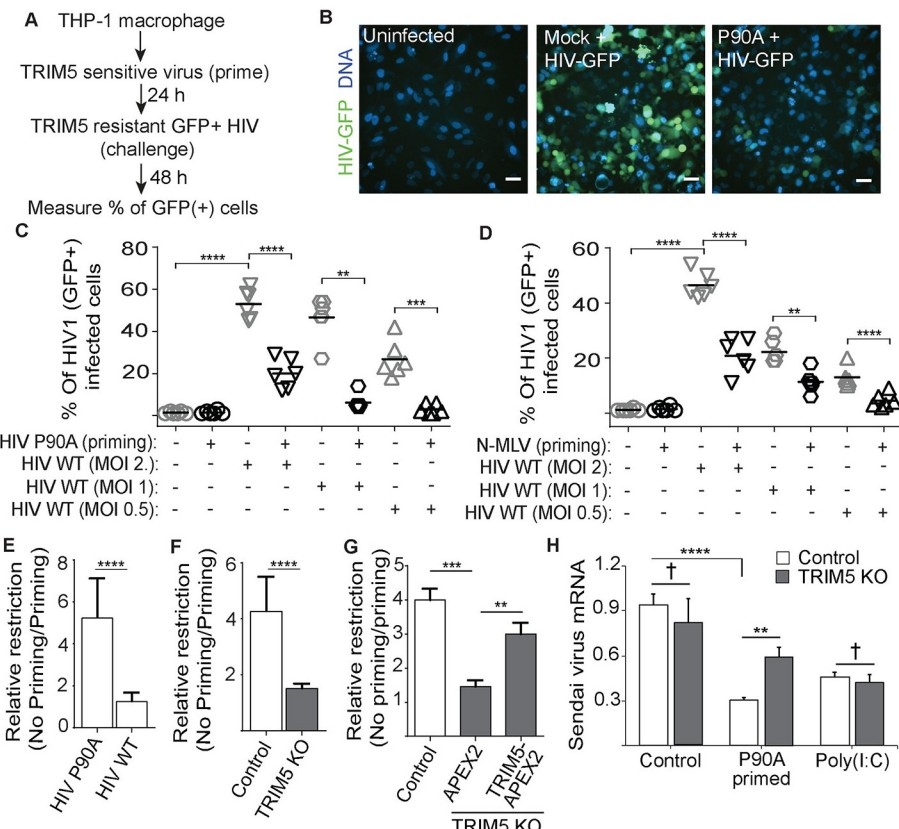

**Fig 6. Stimulation of TRIM5 signaling induces an antiviral state.** (**A**) Schematic overview of experimental model for sequential viral "priming and challenge" experiments. PMA-differentiated THP-1 macrophages were cultured in the presence of the "priming" virus for 24 hours prior to subsequent "challenge" with GFP-expressing WT HIV-1 for next 48h. The percentage of cells showing GFP-positivity was determined by high content imaging. (**B,C**) THP-1 cells were primed with HIV-1 CA P90A (CrFK MOI = 3) prior to challenge with different CrFK MOI of WT HIV-1. Representative images (B) of mock infected, infected, or primed and infected cells captured by high content imager. Scale bar, 50 μm. Plot (C) shows the percentage of cells that become GFP-positive following infection with the challenge virus. (**D**) THP-1 cells were primed with VSV-G pseudotyped N-MLV (CrFK MOI = 0.5) and then challenged with WT HIV-1 at the indicated MOI. For C and D, each data point represents one well of a 96-well plate with >2000 cells analyzed per well. Data shown is from one representative experiment of 3. See also S4A and S4B Fig. (**E**) Differentiated THP-1 cells were primed or not with HIV-1 CA WT (TRIM5 resistant) or P90A (TRIM5 sensitive) and then challenged with HIV-1 GFP. The data are expressed as the ratio of infected cells before priming to relative to what is seen in unprimed cells (relative restriction). (**F**) The effect of TRIM5 knockout on the ability of priming with HIV-1 CA P90A to protect THP-1 macrophages from infection with WT HIV-1 GFP. (**G**) The relative restriction seen in P90A primed WT or TRIM5 KO THP-1 macrophages stably expressing APEX2 or TRIM5-APEX2 (TRIM5 addback). For panels E-G, priming virus was used at a CrFK MOI of 3 and HIV-GFP infections were performed at a CrFK MOI of 1. (**H**) Q-RT PCR analysis of the impact of P90A (CrFK MOI = 3) or poly(I:C) priming on the permissivity of WT or TRIM5 KO THP-1 macrophages to Sendai virus infection. **, P < 0.01; ***, P < 0.001, ****, P < 0.0001; †, not significant by t test or ANOVA.

two different modules of the autophagy machinery (ULK1 and BECN1) were essential for the broadening of TRIM5's antiviral efficacy seen in STR.

## Autophagy factors scaffold TRIM5 interactions with TAK1 and promote TAK1 activation

We next investigated the mechanism underlying how autophagy factors facilitate TRIM5-dependent signaling. The kinase TAK1 is reported to be activated immediately downstream of TRIM5 in response to TRIM5-generated K63-linked poly-ubiquitin chains [12]. This triggers

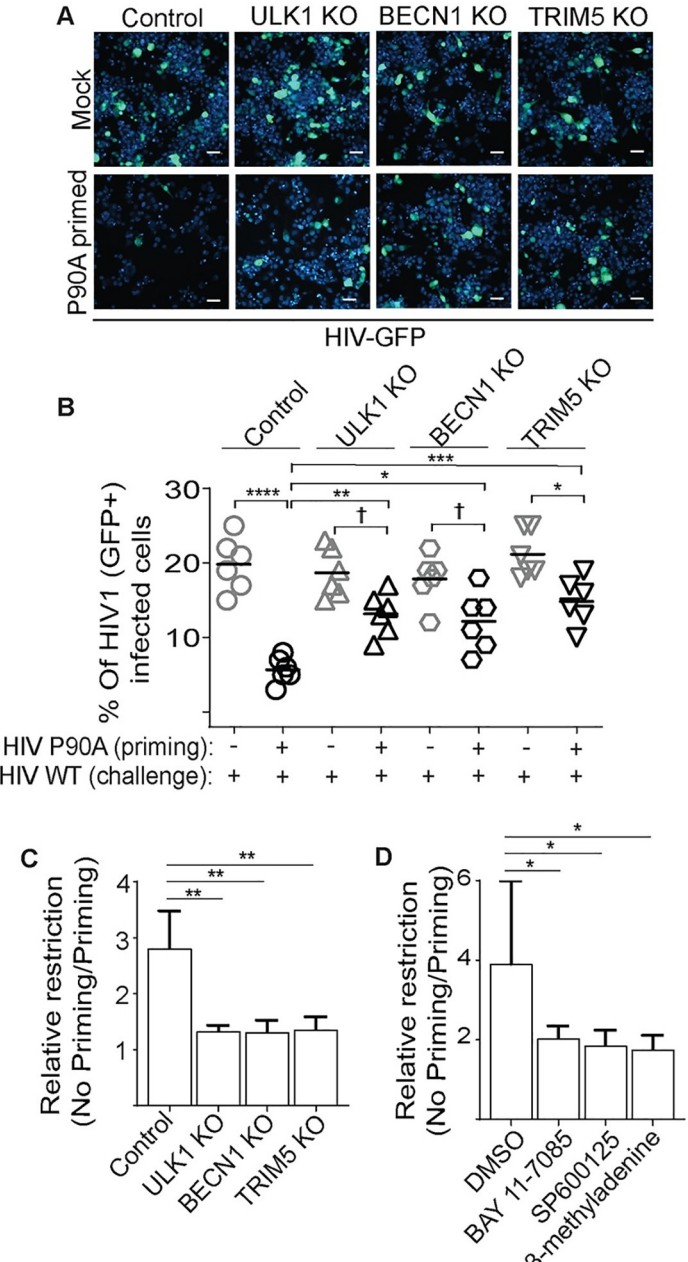

**Fig 7. The establishment of a TRIM5-dependent antiviral state requires autophagy factors.** Control or knockout THP-1 macrophages were primed with P90A (CrFK MOI 3) for 24 hours prior to infection with WT HIV-GFP (CrFK MOI 1) for two days. (**A**) Representative images from high content imager showing HIV-GFP exposed primed or unprimed THP-1 cell lines. Scale bar, 50 μm. (**B**) The percentage of cells showing GFP-positivity. Each data point represents one well of a 96 well plate (>2000 cells analyzed). Data shown is from one representative experiment. (**C**) The relative restriction imposed by P90A priming in control THP-1 macrophages or in THP-1 cell lines knocked out for the indicated genes as shown in (A,B). N = 4 independent experiments. (**D**) The impact of NF-κB, AP1, and autophagy inhibition on P90A-induced restriction. THP-1 cells were treated with BAY 117085 (NF-κB inhibitor; 10 μM), SP600125 (JNK/AP1 inhibitor; 10 μM) or 3-methyladenine (PI3 kinase/autophagy inhibitor; 1 mM) or DMSO (vehicle control) for 1 h prior to infection with P90A or mock infection. See also S4G Fig. Data, mean + SEM. N = 4 independent experiments. *, P < 0.05; **, P < 0.01; ***, P < 0.001; ****, P < 0.0001; †, not significant by ANOVA.

a phospho-relay cascade leading to the activation of Iκκ kinases which subsequently phosphorylate the NF-κB inhibitor IκB, leading to IκB's proteasomal degradation and NF-κB activation. Additionally, TAK1 can activate other MAP kinases ultimately resulting in the phosphorylation and activation of AP1 transcription factors C-Fos and C-Jun. We have found that autophagy factors impact the outcome of these TRIM5-driven responses, with their depletion attenuating anti-retroviral responses. Because autophagy factor knockdowns impact both NF-κB and AP1 signaling downstream of TRIM5 expression, we hypothesized that autophagy factors are acting on the level of TAK1. Accordingly, TRIM5 expression induced the abundance of the active phosphorylated form of TAK1 (pThr184/187) in wild type but not BECN1 or ATG7 knockout cells (Fig 8A). We and others have previously shown that TRIM5 and the TAK1 complex (TAK1, TAB2/3) share a number of binding partners including p62 and BECN1 [16,24,32,33]. Furthermore, we found that TRIM5 and TAK1 colocalize with the autophagy factors p62 and ULK1 in cytoplasmic bodies (Fig 8B and 8C). These shared interactions between TRIM5, TAK1, and the autophagy machinery suggest a model in which the autophagy machinery may provide a scaffold assembling TRIM5 and TAK1. We tested this model by determining the ability of TRIM5 and TAK1 to co-immunoprecipitate in wild type and autophagy-factor knockout HEK293T cells (Fig 8D and 8E). We found that TRIM5-TAK1 interactions were reduced in cells lacking BECN1 and ATG7. Moreover, we observed that TAK1 interactions with ubiquitin, which are required for TAK1 activity, are reduced in GFP-TRIM5-expressing HEK293T cells lacking BECN1, ATG7, and ULK1 (S5A Fig). Together, our data indicate that these autophagy factors facilitate the interaction between TRIM5 and TAK1, and that without this interaction TRIM5 cannot stimulate TAK1 activation.

We next considered a model in which the autophagy machinery helps scaffold TRIM5-TAK1 interactions. A prediction of this model would be that disrupting TRIM5's ability to interact with the autophagy machinery would impact TRIM5 signaling. We previously identified two nearly adjacent LC3-interacting motifs (LIRs) in the TRIM5 coiled-coil domain: 187FEQL190 (LIR1) and 196WEESN200 (LIR2) [16]. Simultaneous mutation of both of these motifs (ΔLIR1/2) abrogated TRIM5 interactions with GABARAP and other mAtg8s in vitro and in vivo [16,23]. These mutations also alter TRIM5's interactions with additional autophagy factors favoring TRIM5-BECN1 interactions and disrupting TRIM5-ULK1 interactions (S5B and S5C Fig). Interestingly, ΔLIR1/2 TRIM5 is dramatically reduced in its ability to interact with TAK1 in coimmunoprecipitation experiments (S5D Fig) and to promote TAK1 activation as measured using phospho-specific antibodies (S5E Fig). Accordingly, GFP-TRIM5 ΔLIR1/2 is attenuated in its NF-κB and AP1 activating potential in reporter cell assays (S5F Fig). These studies indicate that TRIM5 interactions with the autophagy machinery, mediated by its LIRs, are important for its ability to signal through TAK1. However, an important caveat to these studies is that mutations of some residues found in LIR1, notably F187A, can disrupt TRIM5 self-assembly and retroviral restriction [34]. Thus, our data show an involvement of the LIRs and/or of TRIM5 higher-order structures in driving TAK1 activation.

To alleviate this ambiguity, we carried out experiments using LIR2-mutated TRIM5 (196WE ➔ AA; ΔLIR2) [16]. Another group independently identified this motif as being important for mAtg8 binding but showed that LIR2 mutations did not affect TRIM5-depedent CSR [23], suggesting that these mutations do not grossly impact TRIM5 self-assembly. Nevertheless, we found that the ability of the LIR2 mutant of TRIM5 to increase TAK1 activation and to trigger NF-κB activation was reduced relative to what is seen with WT TRIM5 (Fig 8F and 8G). In conclusion, these data demonstrate an essential role for autophagy factors in assembling TRIM5-TAK1 complexes and in TRIM5-dependent TAK1 activation. TRIM5 mutations that have altered interactions with the autophagy machinery are attenuated in their

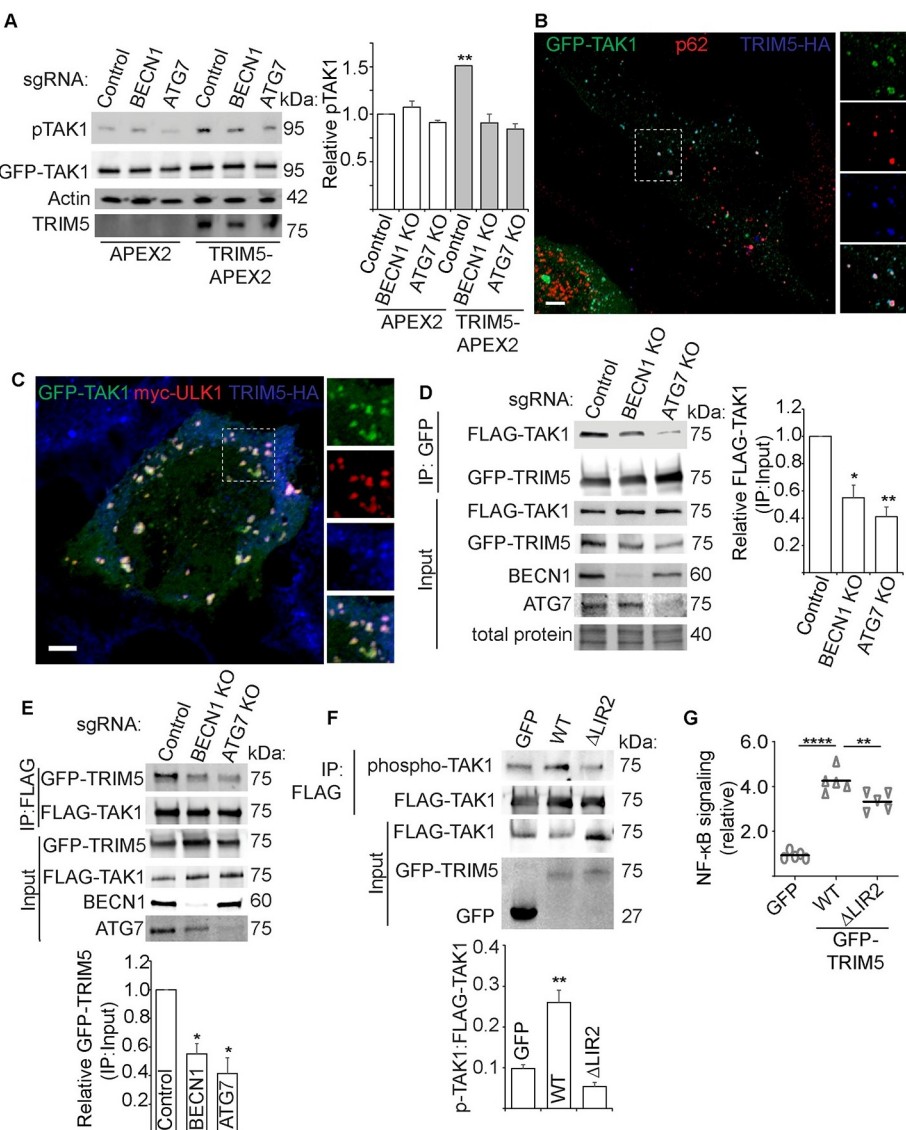

**Fig 8. Autophagy factors scaffold TRIM5 interactions with TAK1 and promote TAK1 activation.** (**A**) Immunoblot analysis of the impact of TRIM5 expression on TAK1 activation in control and autophagy factor knockout HEK293T cells. Lysates from cells transiently transfected with GFP-TAK1 and either APEX2-alone or TRIM5-APEX2 were immunoblotted and probed with the indicated antibodies. Right, the abundance of active phospho-TAK1 (phospho-Thr184/187) relative to total GFP-TAK1. N = 3 experiments. (**B**) Confocal microscopic analysis of GFP-TAK1, TRIM5-HA (rhesus; blue), and endogenous p62 (red) in HeLa cells. Dashed box, region shown in zoomed-in insets. Scale bar, 5 μm. (**C**) Confocal microscopic analysis of GFP-TAK1, Myc-ULK1 (red) in HeLa cells that stably express TRIM5-HA (blue). Dashed box, indicates regions shown in zoomed-in insets (right). Scale bar, 5 μm. (**D, E**) Co-immunoprecipitation analysis of interactions between FLAG-TAK1 and GFP-TRIM5 in WT and autophagy factor KO cells. GFP-TRIM5 was immunoprecipitated in (D) and FLAG-TAK1 was immunoprecipitated in (E). Plots show the abundance of the immunoprecipitated protein (FLAG-TAK1 in D and GFP-TRIM5 in E) relative to that protein's abundance in the input. N = 3 independent experiments. (**F**) Immunoblot analysis of the impact of WT or LIR2 mutant (WE196AA) TRIM5 on the abundance of active phospho-TAK1 (phospho-Thr184/187). Plot shows the abundance of phospho-TAK1 relative to that of FLAG-TAK1. N = 3 independent experiments. (**G**) The impact of WT or LIR2 mutant TRIM5 on the activation of an NF-κB luciferase reporter. Each data point represents an independent biological replicate. *, P < 0.05; **, P < 0.01; ****, P < 0.0001 by ANOVA.

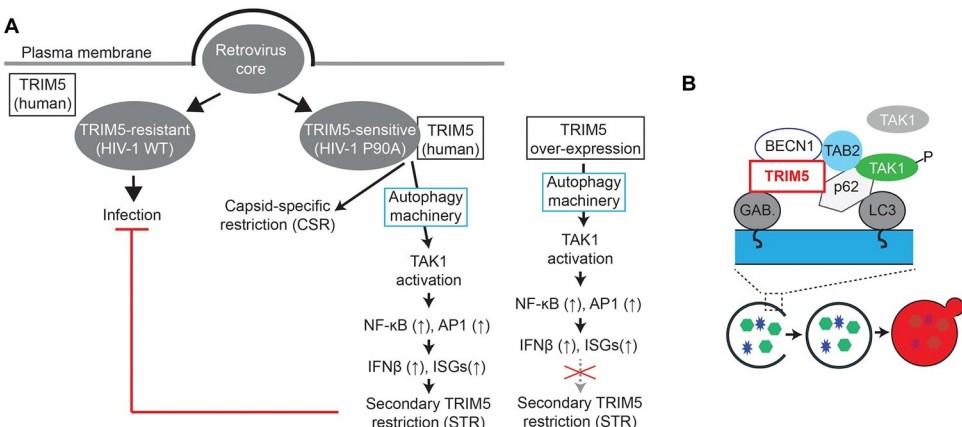

**Fig 9. Model of how autophagy contributes to TRIM5-TAK1 signaling and the establishment of an antiviral state.**
(**A**) TRIM5-sensitive retroviral cores are recognized by TRIM5, leading to direct capsid-specific restriction and triggering a signal transduction cascade that results in the establishment of a general antiviral state. While TRIM5 over-expression can mimic some aspects of viral recognition by TRIM5, it is insufficient to enable viral restriction by TRIM5-resistant viruses. (**B**) Autophagy-dependent assembly of TRIM5-TAK1 complexes that result in TAK1 activation by TRIM5. The various protein-protein interactions depicted are based on findings in this work and in previous reports. Autophagic degradation of negative regulators of TRIM5-TAK1 signaling may also contribute.

ability to carry out signaling functions. These data provide a mechanistic basis for the actions of autophagy factors in TRIM5-based antiviral signaling and STR.

## Discussion

Our study has demonstrated that the autophagy machinery is essential for TRIM5 to carry out its function as a pattern recognition receptor for retroviral cores (Fig 9A). As evidence for this, TRIM5-induced activation of pro-inflammatory transcription factors NF-κB and AP1 or expression of their target genes (e.g. interferon β, IL-6, and NLRP1) is lost in cells lacking components of the autophagy machinery (Figs 1–4). As a pattern recognition receptor, TRIM5 can stimulate the expression of inflammatory and antiviral gene products in response to retroviral detection [12–14]. The ability of TRIM5 to transduce antiviral signaling is dependent on its capacity to interact with the intact retroviral core; hence human TRIM5 is activated by N-MLV but not HIV-1. In this study, we showed that an HIV-1 capsid mutant that was recently demonstrated to be bound and restricted by human TRIM5 [29,30] could activate TRIM5-dependent signaling in macrophage-like cells. Importantly, the ability of this virus to stimulate the expression of the key antiviral cytokine interferon β was lost in autophagy-deficient cells, again demonstrating the importance of the autophagy machinery in TRIM5 signaling (Fig 5).

We found that the signaling induced by HIV-1 CA P90A infection could potently inhibit the ability of otherwise TRIM5-resistant HIV-1 and Sendai virus to infect THP-1 macrophages. Importantly, the antiviral state established in response to HIV-1 CA P90A treatment required TRIM5 and the autophagy factors BECN1 and ULK1 (Figs 6 and 7). We refer to this more broadly antiviral activity of TRIM5 as "secondary TRIM5 restriction" (STR) because it is secondary to priming by a virus subject to capsid-specific restriction (e.g. HIV-1 CA P90A). One important question for future study would be to uncover the mechanisms underlying STR. For example, it remains unknown if the role of TRIM5 and the autophagy machinery is primarily in activation of innate immune signaling in response to priming by viral capsid or whether they also have effector roles in elimination of the super-infecting "challenge" virus.

Another important question is why over-expression of human TRIM5, which enhances IFN-β expression, is insufficient to recapitulate the viral restriction seen in our STR experiments (Fig 9A). We speculate that TRIM5 over-expression is an imperfect surrogate for the TRIM5-dependent cellular responses to viral capsid binding, and thus fails to activate the same antiviral effectors as does *bona fide* TRIM5 activation.

While our STR studies were not intended to mimic a clinically relevant setting, our results may provide an answer as to how human TRIM5, which has historically been considered ineffective as an HIV-1 restriction factor, can confer protection against HIV-1 infection risk in people [9–11]. The P90A capsid mutation is only different from the WT by a single amino acid substitution, yet this change dramatically impacts the virus' susceptibility to TRIM5 CSR and its ability to prime STR. Given the notoriously high mutation rate of HIV-1, it would be surprising if natural transmission did not include some virions with TRIM5-detected capsids capable of stimulating STR. It is possible that the STR triggered by these TRIM5-restricted virions could be sufficient to prevent successful HIV-1 transmission in some cases.

How does the autophagy machinery impact TRIM5 signaling? The loss of autophagy proteins impacts both TRIM5-mediated NF-κB and AP1 activation, and thus we reasoned that autophagy proteins must be acting on the level of TAK1 since this kinase functions as a signaling node controlling both of these transcription factors. Accordingly, our studies demonstrated that the loss of autophagy factors ATG7, BECN1 or ULK1 reduced the ability of TRIM5 to promote TAK1 activation (Fig 8). TAK1 has several functional overlaps with another kinase (receptor-interacting serine/threonine protein kinase 1, RIPK1) that has been proposed to require a subset of autophagy proteins to provide a structural platform for activation [35]. We thus asked whether the autophagy machinery might provide a foundation upon which TRIM5 can interact with and activate TAK1. TRIM5 and TAK1 share a number of autophagy-related interacting partners [16,24,32,33] and we found that TRIM5 and TAK1 co-localize to punctate structures with autophagy factors. Importantly, autophagy factor knockout substantially reduced interactions between TRIM5 and TAK1 (Fig 8), thus demonstrating that the autophagy machinery helps bring TRIM5 and TAK1 together allowing for TRIM5-dependent TAK1 activation. The effects of autophagy factor depletion on TAK1 activation and signaling are recapitulated with TRIM5 mutants in which the LC3-interacting regions (LIRs) have been disrupted resulting in reduced interactions with mAtg8 proteins (LC3s and GABARAPs). Collectively, the data described above support a model in which the autophagy machinery provides a platform upon which TRIM5 can interact with and activate TAK1 (Fig 9B). This non-canonical role for autophagy in TRIM5-TAK1 signaling likely involves multiple protein-protein and/or protein-membrane interactions.

One surprising finding of our study was that all stages of the autophagy pathway, including the later degradative steps, appear to be important for TRIM5-dependent signaling. We used luciferase expression reporter assays to screen different modules of the autophagy machinery for roles in TRIM5-induced signaling. We found that factors acting in autophagy initiation (ULK1 and BECN1), substrate selectivity (p62), autophagosome membrane elongation (ATG7 and GABARAP), autophagosome closure (CHMP2A), and autophagosome-lysosome fusion (VAMP8 and SNAP29) are all important for NF-κB and AP1 transcriptional activation in cells transiently expressing TRIM5. These findings strongly suggest that the TRIM5-TAK1 signaling axis requires full autophagy rather than single components of the autophagy machinery to assemble TRIM5-TAK1 complexes. However, why autophagosome-lysosome fusion mediated by VAMP8 and SNAP29 is important for TRIM5 signaling is still an open question. Our data indicated that inhibition of lysosomal proteases mildly attenuated the ability of TRIM5 expression to stimulate NF-κB activation, thus one possible answer could be that autophagy degrades

or deactivates an as-yet unidentified negative regulator or TRIM5-TAK1 signaling; with one or more of the ~100 deubiquitinating enzymes being likely candidates.

In conclusion, our study shows that the autophagy machinery provides a signaling scaffold allowing TRIM5 to initiate an antiviral state that is protective against HIV-1. This explains the extensive connections between TRIM5 and autophagy and establishes a novel non-canonical role for the autophagy machinery in assembling TRIM5-TAK1 complexes. Finally, TRIM5 is one member of the large TRIM protein family. A very high percentage of the ~70 TRIMs in human genome act in regulating inflammatory signaling and antiviral defense and are increasingly linked to autophagy [22]. Our findings presented here could potentially provide a groundwork for understanding the molecular underpinnings of these other TRIMs and how they coordinate their activities in autophagy and inflammation.

## Materials and methods

### Cell culture

HEK293T, HeLa, THP-1 and CrFK cells were obtained from the American Type Culture Collection (ATCC). HEK293T, HeLa and CrFK cells were grown in Dulbecco's modified Eagle's medium (Life technologies,11965126) supplemented with 10% fetal bovine serum (FBS, Life technologies, 26140–079), 100 U/ml penicillin and 100 μg/ml streptomycin at 37˚C in a 5% CO2 atmosphere. HeLa cells stably expressing HA-tagged RhTRIM5α were obtained from NIH AIDS reagents and were maintained in the above media supplemented with 1 μg/ml puromycin. THP-1 cells were maintained in RPMI 1640 (Corning, 10-040-CV) containing 2 mM L-glutamine, 1.5 g/l sodium bicarbonate, 4.5 g/l glucose (Corning, 25-037-CL), 10 mM HEPES (ThermoFisher, 15630080) and 1.0 mM sodium pyruvate (ThermoFisher, 11360–070), 10% FBS, 100 U/ml penicillin and 100 μg/ml streptomycin. THP-1 cells were differentiated with 50 ng/mL PMA (Sigma, P8139) for 24h, washed and incubated in complete medium for 48h before experimental use. APEX2-V5, RhTRIM5-APEX2-V5 and HuTRIM5-APEX2-V5 stable overexpression in WT or KO THP-1 backgrounds was achieved by viral transduction followed by 14–21 days of culturing in medium containing the selective antibiotic (1 μg/ml puromycin) before stable integration of the target gene was confirmed by western blotting. ULK1, BECN1, ATG7 and TRIM5 knockout (KO) HEK293T and THP1 cells were generated by transduction with lentiCRISPRv2-based lentiviruses followed by 2–4 weeks of culturing in medium containing the selectable marker (1 μg/ml puromycin for ULK1, BECN1, ATG7 KO cells or 200 μg/ml hygromycin for TRIM5 KO cells). Knockout lines were confirmed by immunoblot.

### Generation of single-cycle lentivirus or retrovirus for stable cell line construction or infection assays

Viral particles for the generation of stable overexpressed cell lines were produced by co-transfection of pLEX_307 (a gift from David Root, Addgene plasmid # 41392) containing the target gene, psPAX2 and pMD2.G at the ratio of 1:1:1 in HEK293T cells using ProFection Mammalian Transfection System (Promega, E1200), medium was changed 16h post transfection and virus containing supernatant was harvested 48h later, clarified by centrifuging for 5 min at 1200 rpm, 0.45 μm-filtered (Millipore, SE1M003M00), diluted with full medium at 1:1 ratio and used to transduce target cells for 48h in 6 cm dishes.

Viral particles for the generation of knockout cell lines were produced by transfecting HEK293T cells with a lentiviral vector, lentiCRISPRv2 carrying both Cas9 enzyme and a guide RNA targeting specific gene together with the packaging plasmids psPAX2 and pMD2.G at the

ratio of 10 μg, 10 μg and 10 μg/10 cm dish. Lentiviral particles were harvested from supernatants as mentioned above and HEK293T, Huh7, and THP-1 cells were infected in the presence of polybrene for 48h in 6 cm dishes.

In this study, all virus infection was performed using VSV-G pseudotyped single-cycle HIV-1 or N-MLV produced by transfection of HEK293T cells. 24h before transfection, $2.2\times10^6$ HEK293T cells were seeded in 10 cm dishes. Two-part single cycle VSV-G-pseudotyped HIV-1 (NL43 strain) was collected from the supernatants of HEK293T cells transfected with plasmids encoding VSV-G and HIV-1 lacking the Env gene (Env-, VPR-, Nef+, IRES-GFP) at 1:2 ratios. Three part, single cycle VSV-G-pseudotyped N-MLV, HIV1 WT and recombinant HIV-1 strain (HIV-1 CA P90A) were produced by co-transfecting 15 μg lentivector genome plasmid (pLVX-mcherry), 10 μg gag-pol plasmid (pCIG3-N for N-MLV, WT and P90A HIV-1 capsid[29]; kindly provided by Edward Campbell) and 5 μg VSV-G plasmid. 16h post transfection, the culture media was changed, viral supernatant was harvested at 72h, clarified by centrifuging for 5 min at 1200 rpm, passed through a 0.45 μm filter and stored at -80˚C. The multiplicity of infection (MOI) of different virus stocks used in this study was determined by infecting permissive, TRIM5-null, 8000 CrFK cells with serial dilutions of virus and identifying the $ID_{50}$ by high content imaging. This value was then back-calculated to determine the CrFK MOI.

## Plasmids, siRNA, and transfection

pDest40-APEX2-V5 and pLEX_307-APEX2-V5; pDest40-RhTRIM5-APEX2-V5 and pLEX_307- RhTRIM5-APEX2-V5; pDest40-HuTRIM5-APEX2-V5 and pLEX_307-HuTRIM5-APEX2-V5; LIR2 mutant pDest40-GFP-RhTRIM5 were generated using Gateway recombination cloning. First, they were PCR amplified from available cDNA clones and recombined into pDONR221 using the BP reaction (Life Technologies, 11789–020) prior to being recombined into expression plasmids by LR cloning (Life Technologies, 11791–020). Plasmid constructs were verified by DNA sequencing. The AP1 luciferase reporter plasmid was a gift from Alexander Dent (Addgene plasmid #40342; 3XAP1pGL3), the NF-κB luciferase reporter was purchased from Promega (#E8491) and the Renilla luciferase plasmid (pRL-SV40, Addgene plasmid #27163) was a gift from Ron Prywes. All other plasmids have been previously published [16,32]. All siRNA smart pools were from Dharmacon. siRNA was delivered to cells using Lipofectamine RNAiMAX (ThermoFisher, 13778150). Plasmid transfections were performed using Lipofectamine 2000 (ThermoFisher, 11668019) or Calcium Phosphate (Promega, E1200). Samples were prepared for analysis the day after DNA transfection. For siRNA experiments, cells were harvested 48–72h after siRNA transfection.

## Antibodies and reagents

The following primary antibodies were used: Beclin1 (Cell Signaling, 3495S), ATG7 (Cell Signaling, 8558S), ULK1 (Cell Signaling, 8054S), p62 (BD, 610833), UBC13 (Abcam, 25885), GABARAP (Cell Signaling, 13733S), TFEB (Cell Signaling, 37785S), CHMP2A (Proteintech, 10477-1-AP), VAMP8 (Abcam #76021), SNAP29 (Abcam #138500), V5 (Cell Signaling, 13202S), TRIM5 (Abcam, Ab59000; Cell Signaling #14326), phospho-TAK1 (Cell Signaling, 4508S), FLAG (Sigma, F1804), GFP (Abcam, Ab290), HA (Abcam, Ab9110), c-Myc (Santa Cruz, 40) and actin (Santa Cruz, 58673). Secondary antibodies used were fluorescently conjugated goat anti-mouse (LI-COR, 925–68020) and goat anti-rabbit (LI-COR, 925–32210), HRP-conjugated goat anti-mouse (Bio-Rad, 1721011) and goat anti-rabbit (Bio-Rad, 1721019) or Clean-Blot HRP (ThermoFisher, 21230). Nuclease free water (Dharmacon, B-003000-WB-100), 5X siRNA buffer (Dharmacon, B-002000-UB-100), Opti-MEM Reduced Serum Medium

(ThermoFisher, 31985070), RIPA lysis buffer (ThermoFisher, 89901), phenylmethylsulfonyl fluoride (PMSF, Sigma, 93482, 1 mM), protease inhibitor cocktails (ROCHE, 11836170001), BSA (Fisher Scientific, BP 1600–1), Puromycin (Sigma, P8833), Polybrene (EMD Millipore, TR-1003-G, 10 µg/ml), Hygromycin (Corning, 30-240-CR), MG132 (Selleckchem, S21619, 0.2 µM), IP lysis buffer (ThermoFisher, 87788), phosphatase inhibitor cocktails (ROCHE, 04906845001), Dynabeads Protein G (ThermoFisher, 10004D) and Restore PLUS Western blot stripping buffer (ThermoFisher, 46430). NF-κB and AP1 inhibitors (BAY 11–7085 and SP600125) were purchased from Enzo Life Sciences. 3-methyladenine, e64d, and pepstatin A were purchased from Sigma.

## Luciferase assays

For siRNA mediated knockdown experiments, HEK293T cells ($1.2 \times 10^6$ cells/6cm dish) were reverse transfected with non-targeting or the indicated siRNA using Lipofectamine RNAi-MAX, after 24h cells were harvested and reseeded in 96 well plate (20000 cells/well in 100 µl tissue culture medium). For other luciferase experiments, 20000 HEK293T cells were plated in each well of a 96 well plate 24h prior to transfection. Cells were transfected using 0.25 µL Lipo-fectamine 2000 per well, with 10 ng of the *Renilla* luciferase internal control reporter plasmid pRL-TK (thymidine kinase promoter dependent *Renilla* luciferase), 20 ng firefly luciferase experimental reporter plasmid (NF-κB or AP1 response element dependent firefly luciferase) and 25 ng of pDest40 containing control (APEX2 or GFP) or TRIM5 or TAK1 fusions. Each experimental condition was performed in quadruplicate—sextuplicate. 40–48h after transfection, the plate was assayed using the Dual-Glo Luciferase Assay System (Promega, E2920) and read using a Microplate Luminometer (BioTek, SYNERGY HTX Multi-Mode reader). Firefly luciferase readings were normalized to Renilla luciferase readings in each well, and the data are represented as fold-change compared to control cDNA.

## Quantitative RT–PCR

Total RNA was extracted with RNeasy Plus Mini kit (Qiagen, 74104) and first strand cDNA was synthesized using random hexamers as primers (High-Capacity cDNA Reverse Transcriptase Kit, Thermo Fisher, 4368814), in accordance with manufacturer's instructions. qPCR reactions were performed with TaqMan Gene Expression Assays (Thermo Fisher) for, IFNB1 (Hs01077958_s1), NLRP1 (Hs00248187_m1), IL6 (Hs00985639_m1), IL10 (Hs00961622_m1), SEV (Mr04269880_mr), and 18S rRNA (Fn0464250_s1) was used as a house keeping gene for normalization. The qPCR assays were run on the StepOnePlus Real-Time PCR System (Applied Biosystems).

## Infection assay using single-cycle viruses

For CrFK, Huh7, and HeLa cells, 8000 cells were seeded per well in a 96 well plate, 24h prior to virus challenge. Media containing VSV-G pseudotyped lentiviral vectors expressing HIV1 WT or HIV1-P90A or N-MLV was added to infect cells in a total volume of 100 µL in the presence of polybrene. For THP1 cells, 50000 cells were seeded in each well of a 96 well plate in a culture medium containing 50 ng/mL PMA for 24h, then washed and placed in normal medium. After 48h, differentiated THP-1 cells were infected with the virus using MOI specified in figure legends. After infection, cells were first incubated at 4˚C for 1h to allow the virus to bind. Free virus was then removed by washing and cells were incubated in complete medium. 48h post infection, cells were fixed, stained with Hoechst 33342 and the fraction of transduced cells showing fluorescent protein positivity was determined by high content imaging and analysis. Culture supernatant from un-transfected HEK293T cells was diluted in complete media and

used for mock infections. Infection was determined by measuring the percentage of GFP or mCherry positive cells. All the infections were repeated multiple times (>3).

## Sequential retroviral infections assessing secondary TRIM5 restriction

THP1 control and the different THP1 KO cells were seeded into 96 well plates at a density of 50000 cells/well, differentiated with PMA, then primed with VSV-G pseudotyped HIV-mCherry (at a CrFK MOI of 3) or VSV-G pseudotyped HIV-mCherry CA P90A (at a CrFK MOI of 3) or VSV-G pseudotyped N-MLV-mCherry (at a CrFK MOI of 0.5), 24h later media was removed, washed and challenged with different dilutions of VSV-G pseudotyped HIV1-WT-GFP tagged (CrFK MOI of 2 was used as the starting dilution). For most experiments, we challenged cells with a CrFK MOI of 1. 48h post second virus infection, cells were fixed, stained with Hoechst 33342 and percentage of GFP positive cells were determined using high content imaging. Where indicated, cells were treated with 0.2 μM MG132 during the virus priming.

## Sendai virus infections

Sendai virus (formerly Parainfluenza Virus 1, Sendai) was obtained through BEI Resources, NIAID, NIH: NR-3227. Cells were inoculated with virus in RPMI media at a concentration of 50 hemagglutination units (HAU)/mL. To quantify experiments, RNA was collected in Trizol Reagent (Invitrogen catalog# 15596026) and purified with the Zymo Research Direct-zol RNA Miniprep Kit (catalog# R2052). Total RNA concentration was normalized and cDNA synthesis performed with Applied Biosystems High-Capacity cDNA Reverse Transcription Kit (catalog# 43-688-14).

## High content imaging

All high content experiments were performed in 96-well plate format. After the indicated treatments, cells were fixed with 4% paraformaldehyde for 10 min, washed twice with 1X PBS and stained with the nuclear stain Hoechst 33342. Nuclear staining was used for autofocus and to automatically define cellular outlines. High content imaging and analysis were performed using a Cellomics CellInsight CX7 scanner and iDEV software (Thermo); > 2000 cells were analyzed per well, and 10–12 wells of the 96 well plate were analyzed per sample. Transduced cells were automatically identified based on having above background fluorescent protein signal in the nucleus. All data acquisition and analysis were computer driven and independent of human operators.

## Co-immunoprecipitation and immunoblotting

Most immunoprecipitation and immunoblots were as described [16]. For whole cell lysates, cell lysis was performed with a modified RIPA buffer containing 0.5% NP-40, 1% Triton X-100, and 0.5% SDS. For immunoprecipitation, cell lysis was performed with an IP lysis buffer containing 1% NP-40. Lysis buffers contained PMSF, protease inhibitor cocktails and phosphatase inhibitor cocktails. Immunoblots were imaged using a Biorad Chemidoc MP or a Licor Odyssey system.

## Confocal microscopy

HeLa cells stably expressing HA-tagged RhTRIM5α were plated onto glass coverslips in 12 well plates prior to being transfected with the plasmids indicated in figures. Samples were fixed with 4% paraformaldehyde for 10 min, permeabilized with 0.1% saponin in 3% BSA and

blocked with 3% BSA in PBS. Intracellular targets were then stained with primary antibodies according to the manufacturer's recommendation, washed three times with PBS, followed by incubation with Alexa Fluor conjugated secondary antibodies for 1 h at room temperature. Coverslips were mounted using ProLong Diamond Antifade Mountant (Invitrogen, P36970). Images were acquired using a Zeiss LSM800 microscope and analyzed using Zen2 software (Zeiss) and deconvolved using Huygens Essential software (Scientific Volume Imaging).

## Statistical analysis

Data are expressed as means ± SEM (n>3). Data were analyzed with unpaired two-tailed t-tests or ANOVA with Bonferroni post hoc analysis. Statistical significance is defined as *, $P < 0.05$; **, $P < 0.01$; ***, $P < 0.001$; ****, $P < 0.0001$.

## Supporting information

**S1 Fig. Luciferase reporter system reveals role for multiple autophagy factors in TRIM5 signaling.** (**A**) The impact of GFP-TRIM5, TRIM5-APEX2, wt GFP-TAK1 and kinase-dead GFP-TAK1 (K63W) relative to GFP or APEX2 alone on AP1 activity using a dual-luciferase reporter system. (**B**) Dual-luciferase reporter-based assays determining the effects of siRNA-mediated knockdown of UBC13 on the ability of TRIM5 expression to drive activation of NF-κB or AP1. Immunoblots illustrate knockdown efficiency. (**C**) Immunoblots showing knockdown efficiency corresponding to the data sets shown in Fig 3A and 3E. Data,****, $P < 0.0001$ by ANOVA.
(TIF)

**S2 Fig. Expression of human and rhesus TRIM5-APEX2 in THP-1 macrophage-like cells.** (**A**) Immunoblot analysis of THP-1 cells stably transduced with lentiviruses expressing APEX-V5 alone (AP2-V5) or TRIM5-APEX2-V5 of human or rhesus origin. Lysates were harvested from THP-1 cells following culture in media containing selective antibiotic, and immunoblots were probed with anti-V5 or anti-actin. (**B**) Quantitative RT-PCR analysis of IL-10 expression in THP-1 macrophages stably transduced as indicated prior to siRNA-mediated knockdowns of *BECN1* or *ATG7*. NT, non-targeting siRNA. (**C,D**) Quantitative RT-PCR analysis of knockdown efficiency in differentiated THP-1 cells corresponding to the data shown in (B) and in Fig 2D and 2F. N = 3 independent experiments. Data, *, $P < 0.05$; **, $P < 0.01$; ***, $P < 0.001$, **** by ANOVA.
(TIF)

**S3 Fig. HIV-1 P90A capsid, but not WT capsid, induces interferon β expression.** (**A**) Immunoblot analysis of TRIM5 expression in THP-1 cells transduced with lentivirus encoding Cas9 and either non-targeting (control) or two different TRIM5-targetted guide RNAs. (**B**) High content imaging-based analysis of GFP-expressing N-MLV infection of control and TRIM5 knockout THP-1 macrophages. (**C**) Quantitative RT PCR analysis of interferon β mRNA in wild type THP-1 macrophages 2 hours after infection with VSV-G pseudotyped HIV-1 (WT or P90A capsid), both used at a CrFK MOI of 3. TLR-3 ligand poly(I:C) was used as a positive control. (**D**) Immunoblot analysis of autophagy-factor knockout THP-1 cells generated by transduction with a Cas9/guide RNA expressing lentiviral vector. Data, *, $P < 0.05$; **, $P < 0.01$; ***, $P < 0.001$; ****, $P < 0.0001$; †, not significant by ANOVA.
(TIF)

**S4 Fig. Priming cells with TRIM5-restricted retroviruses induces an antiviral state.** (**A**) The relative restriction of the challenge virus imposed by prior infection is calculated by dividing the percent of infected cells without priming by the percent of cells after priming. Data,

mean + S.E.M; N = 3 experiments as shown in Fig 6C in which cells were primed with HIV-1 CA P90A (CrFK MOI 3) and challenged with different dilutions of WT HIV-GFP. (**B**) The relative restriction of HIV-GFP following priming (or not) with N-MLV. Data, mean + S.E.M; N = 3 experiments as shown in Fig 6D. Cells were primed with a CrFK MOI of 0.5. (**C**) The impact of priming with TRIM5-restricted (P90A) or TRIM5-resistant (WT) HIV-1 capsids corresponding to data shown in Fig 6E. Images show representative micrographs from high content imager showing the impacts of priming on HIV-GFP transduction efficiency. Scale bar, 50 μm. (**D**) Immunoblot analysis of TRIM5 knock-out THP-1 cells transduced with lentivirus expressing APEX2-V5 alone or human TRIM5-APEX2-V5 corresponding to Fig 6G. (**E**) STR in Huh7 hepatoma cell line. Immunoblot shows TRIM5 knockout efficiency in Huh7 cells. Plot (right), control or TRIM5 KO Huh7 cells were exposed or not to VSV-G pseudo-typed HIV-1 P90A (CrFK MOI = 0.25) 24 h prior to being challenged by GFP-expressing VSV-G pseudotyped WT HIV-GFP (CrFK MOI = 0.5). Two days later, the percentage of cells showing green fluorescence was determined by high content imaging. (**F**) High content imaging-based analysis of WT HIV-GFP infection of control and ATG7 knockout THP-1 cells without priming. Data shown, 1 of 3 experiments. Each data point represents one well of a 96 well plate with >2000 cells analyzed per well. (**G**) The effect of NF-κB, AP1, and autophagy inhibitory compounds on STR in THP-1 macrophages corresponding to data shown in Fig 7D. Each data point represents one well of a 96 well plate with >2000 cells analyzed per well by high content imaging. (**H**) The effect of proteasome inhibition with MG132 on the ability of HIV-1 CA P90A priming on WT HIV-GFP infection. Cells were treated with MG132 for 1 hour prior to priming (CrFK MOI 3) and cultured in the presence of MG132 for the remainder of the experiment. HIV-GFP was used at a CrFK MOI of 1. Data shown, 1 of 2 experiments. Each data point represents one well of a 96 well plate with >2000 cells analyzed per well. *, $P < 0.05$; **, $P < 0.01$; ***, $P < 0.001$; ****, $P < 0.0001$; †, not significant by ANOVA. (TIF)

**S5 Fig. Impact of LC3-interacting region mutations on TRIM5-TAK1 signaling.** (**A**) Co-immunoprecipitation analysis of the interactions between HA-ubiquitin and FLAG-TAK1 in WT or autophagy factor knockout cells expressing GFP-TRIM5. (**B,C**) Co-immunoprecipitation analysis of interactions between GFP-tagged WT and ΔLIR1/2 mutant (F187A, L190A, W196A, E197A) TRIM5 and FLAG-BECN1 (B) or Myc-ULK1 (C) from transiently transfected HEK293T cell lysates. Anti-GFP was used for pull-down. (**D**) Co-immunoprecipitation analysis of interactions between GFP-tagged WT or ΔLIR1/2 TRIM5 and FLAG-TAK1 from lysates of transiently transfected HEK293T cells subjected to immunoprecipitation with anti-FLAG (top) or anti-GFP (bottom) with immunoblots probed as indicated. (**E**) The effect of WT or ΔLIR1/2 GFP-TRIM5 on the abundance of active phospho-TAK1 (phospho-Thr184/187) in transiently transfected HEK293T cell lysates. (**F**) The impact of ΔLIR1/2 mutation on the ability of GFP-TRIM5 to induce NF-κB and AP1 activation as measured by luciferase reporters in HEK293T cells. Each data point represents an independent biological replicate, ***, $P < 0.001$; ****, $P < 0.0001$ by ANOVA. (TIF)

## Acknowledgments

We thank Dr. Edward Campbell (Loyola University Chicago) for providing plasmids and for helpful discussions. We thank Dr. Kiran Bhaskar (University of New Mexico) for the use of his qPCR instrument and Drs. Diane Lidke and Suresh Kumar (University of New Mexico) for critically reading the manuscript.

## Author Contributions

**Conceptualization:** Michael A. Mandell.

**Funding acquisition:** Michael A. Mandell.

**Investigation:** Bhaskar Saha, Devon Chisholm, Alison M. Kell, Michael A. Mandell.

**Methodology:** Michael A. Mandell.

**Supervision:** Michael A. Mandell.

**Writing – original draft:** Bhaskar Saha, Michael A. Mandell.

**Writing – review & editing:** Bhaskar Saha, Alison M. Kell, Michael A. Mandell.

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
