## [Decision Letter · Decision Letter 0]

11 Aug 2020

Dear Dr. Mandell,

Thank you very much for submitting your manuscript "A non-canonical role for the autophagy machinery in anti-retroviral signaling mediated by TRIM5α." for consideration at PLOS Pathogens. As with all papers reviewed by the journal, your manuscript was reviewed by members of the editorial board and by several independent reviewers. In light of the reviews (below this email), we would like to invite the resubmission of a significantly-revised version that takes into account the reviewers' comments. In particular, we would need to see a meaningful response to the comments of reviewer 1 and some rewriting of the results section in response to reviewer 2.

We cannot make any decision about publication until we have seen the revised manuscript and your response to the reviewers' comments. Your revised manuscript is also likely to be sent to reviewers for further evaluation.

Sincerely,

Bryan R. Cullen

Associate Editor

PLOS Pathogens

Susan Ross

Section Editor

PLOS Pathogens

Kasturi Haldar

Editor-in-Chief

PLOS Pathogens

orcid.org/0000-0001-5065-158X

Michael Malim

Editor-in-Chief

PLOS Pathogens

orcid.org/0000-0002-7699-2064

Reviewer's Responses to Questions

**Part I - Summary**

Reviewer #1: TRIM5a is one of the best-studied Tripartite motif proteins. It is well-recognized as a Lentiviral restriction factor. While it is known that TRIM5a recognizes capsids in a species-specific manner, and induces premature uncoating, the consequences and mode of signaling induced by TRIM5a are less clear. Particularly the involvement of autophagy has been under debate.

It has been proposed that TRIM5a may serves as an innate immune receptor. Although the pattern recognized by this factor may be really specialized to qualify as a pattern recognition receptor.

In this manuscript Saha and co-workers show that TRIM5-dependent cytokine/Nf-kB induction requires core autophagy proteins such as ATG7 or beclin-1. Pre-stimulation of cells with TRIM5a-recognized capsid renders the cells less susceptible towards a subsequent HIV-1 infection.

Most experiments are well conducted and controlled. The findings presented would be novel and an interesting aspect of signaling regulation by autophagy. However, there are a few things which may need some clarification.

Reviewer #2: This paper describes a non-canonical role for the cellular autophagy machinery in TRIM5 mediated innate signalling. I think this is a very important result. However, the results section of the paper is at times rather hard to follow with some data seemingly randomly assigned to Figs or Suppl Figs- did a limit in the number of figures play a role? Why for example did parts A and B of Fig 2 appear there when the experiments reported were performed in HEK cells, not macrophages as in the figure title. I wonder whether some of the normalized data would be better presented as tables. I would also note that improved quantification would be very helpful in assessing many of the protein blots.

Reviewer #3: This is an nice study that is comprehensive and well controlled. The paper is well-written and makes important conceptual advances in the field of antiviral signaling pathways, by enhancing our understanding of how autophagy contributes to the ability of TRIM5a to promote the IFN response.

In my opinion, this work will be of general interest to a broad audience in molecular biology and cell biology.

**Part II – Major Issues: Key Experiments Required for Acceptance**

Reviewer #1: - There is a control missing in P90A capsid pre-stimulation assay. How do capsids modulate the IFN-b response and a subsequent infection that are not recognized by TRIM5a? Please include priming with a non TRIM5-sensitive virus in Fig. 4.

- I wonder whether the impact on Nf-kB signaling is a common feature of that signaling pathways or unique for TRIM5a-dependent induction? The authors should include controls in the assays in Fig.1, e.g. overexpression of IKKa/b or other means to stimulate an Nf-kB response?

- In Fig. 2D, the IFNb levels induced by RhTRIM5 and HuTRIM5 are the same, however huTRIM5 has no effect on virus replication. Would that mean that the signaling of TRIM5a does not impact the virus?

- Please include TRIM5a KO in Fig. 3B and C.

- Fig. 6D and E: The IP levels of TAK1 and phosphor-TAK1 visually correlate with Input FLAG-TAK1 or GFP-TRIM5, please quantify these IPs and normalize to the respective controls, to strengthen the conclusion.

Reviewer #2: (No Response)

Reviewer #3: I offer a few minor suggestions for the authors to consider:

1) Potential candidates for the regulation of TAK1 activity by TRIM5a could be TAB2/TAB3. The authors should test in their experimental settings if TAB2/TAB3 levels and/or their interaction with TAK1 are altered by downregulating the expression of autophagy genes.

2) It remains unclear if, besides autophagy proteins, the autophagic degradative activity is also required for the regulation of NF-kB signaling. It would be interesting to test if lysosomal inhibitors affect IFN induction by TRIM5a

**Part III – Minor Issues: Editorial and Data Presentation Modifications**

Reviewer #1: - Please replace Fig. 2C with a more quantitative FACS assay

- Fig 5 C and D would benefit from a more transparent presentation of the data

- Fig 6 A has to be quantified: phosphor-Tak vs Tak levels

- Fig 6B and C, all the single overexpressions and stains need to be shown. Please quantify the co-localisation.

- Fig. 6G: This dataset is relatively weak, although it is explained why. However, are there any other mutants of TRIM5a which may be more suitable to show that autophagy is required for Nf-kB signaling?

- Processing of signaling compounds may be important. But also degradation of NIK (and other negative regulators of Nf-kB signaling) as a major factor (besides DUBs), could be experimentally checked or addressed in the discussion.

- Can the authors include a model of their suggested mechanism?

Reviewer #2: L139ff. It appears that BECN1, ATG7, ULK1 knockdowns/outs have a greater effect on NF-KB than AP-1. Is this true? If so what does it mean?

L141. Requiring

L183ff. There appear to be differences between hu and rhTRIM5, especially in NLRP1 responses. Do you think these are important? I wonder whether it would be worth testing RBCC/PrySpry chimeras?

L202ff. Is over-expression in some way similar to CA binding? Could it be that hexameric TRIM5 is needed for TAK1 activation and that this occurs at a certain rate naturally and is enhanced by over-expression or virus binding?

No reference in text to Fig S3A

L455ff. Is the preparation of deltaLIR1/2 described here?

Fig 2. What is Scr siRNA? Is Beclin1 siRNA the same as BECN1 siRNA in Fig 1?

Fig 4. In Fig 3A CrFK moi-3 gave c.30% infected cells in 4C CrFK a moi of 2 gave c.50% infection. Needs an explanation. Also the differences going from moi of 2 to 1 in Figs 4C,D

Fig 6A. Does ATG7 really increase amount of p-TAK1 in GFP control?

Fig 6B, C. Very hard to make out important details

Fig 6E. Is GFP-TRIM supposed to go down in IP but not in input?

Fig 6F. Why are there two sets of FLAG-TAK1 samples?

Fig S3A. Is it fair to say that on a per cell basis HIV P90A is more efficient that poly(I:C)?

Fig S5B. See labels GFP-RhTRIM5 and GFP-TRIM5. Is the Rh a mistake? Are all these experiments done with wt and deleted human TRIM5?

Reviewer #3: None

PLOS authors have the option to publish the peer review history of their article (what does this mean?). If published, this will include your full peer review and any attached files.

Reviewer #1: No

Reviewer #2: No

Reviewer #3: No
---

## [Editor Report · Decision Letter 1]

1 Oct 2020

Dear Dr. Mandell,

We are pleased to inform you that your manuscript 'A non-canonical role for the autophagy machinery in anti-retroviral signaling mediated by TRIM5α.' has been provisionally accepted for publication in PLOS Pathogens.

Best regards,

Bryan R. Cullen

Associate Editor

PLOS Pathogens

Susan Ross

Section Editor

PLOS Pathogens

Kasturi Haldar

Editor-in-Chief

PLOS Pathogens

orcid.org/0000-0001-5065-158X

Michael Malim

Editor-in-Chief

PLOS Pathogens

orcid.org/0000-0002-7699-2064
---

## [Editor Report · Acceptance letter]

9 Oct 2020

Dear Dr. Mandell,

We are delighted to inform you that your manuscript, "A non-canonical role for the autophagy machinery in anti-retroviral signaling mediated by TRIM5α.," has been formally accepted for publication in PLOS Pathogens.

Best regards,

Kasturi Haldar

Editor-in-Chief

PLOS Pathogens

orcid.org/0000-0001-5065-158X

Michael Malim

Editor-in-Chief

PLOS Pathogens

orcid.org/0000-0002-7699-2064